# Pharmacological Modulation of SAMHD1 Activity by CDK4/6 Inhibitors Improves Anticancer Therapy

**DOI:** 10.3390/cancers12030713

**Published:** 2020-03-18

**Authors:** Marc Castellví, Eudald Felip, Ifeanyi Jude Ezeonwumelu, Roger Badia, Edurne Garcia-Vidal, Maria Pujantell, Lucía Gutiérrez-Chamorro, Iris Teruel, Anna Martínez-Cardús, Bonaventura Clotet, Eva Riveira-Muñoz, Mireia Margelí, Ester Ballana

**Affiliations:** 1IrsiCaixa AIDS Research Institute, Badalona, 08916 Catalonia, Spain; mcnadal89@gmail.com (M.C.); iezeonwumelu@irsicaixa.es (I.J.E.); rbadia@irsicaixa.es (R.B.); egvidal@irsicaixa.es (E.G.-V.); mpujantell@irsicaixa.es (M.P.); lgutierrez@irsicaixa.es (L.G.-C.); bclotet@irsicaixa.es (B.C.); eriveira@irsicaixa.es (E.R.-M.); 2Health Research Institute Germans Trias i Pujol (IGTP), Hospital Germans Trias i Pujol, Universitat Autònoma de Barcelona, 08916 Badalona, Spain; efelip@irsicaixa.es (E.F.); iteruel@iconcologia.net (I.T.); amartinezc@igtp.cat (A.M.-C.); mmargeli@iconcologia.net (M.M.); 3Institut Català d’Oncologia, Badalona, 08916 Catalonia, Spain

**Keywords:** SAMHD1, antimetabolite, CDK4/6, pemetrexed, HIV, cancer

## Abstract

Sterile alpha motif and histidine-aspartic acid domain-containing protein 1 (SAMHD1) is a dNTP triphosphohydrolase involved in the regulation of the intracellular dNTP pool, linked to viral restriction, cancer development and autoimmune disorders. SAMHD1 function is regulated by phosphorylation through a mechanism controlled by cyclin-dependent kinases and tightly linked to cell cycle progression. Recently, SAMHD1 has been shown to decrease the efficacy of nucleotide analogs used as chemotherapeutic drugs. Here, we demonstrate that SAMHD1 can enhance or decrease the efficacy of various classes of anticancer drug, including nucleotide analogues, but also anti-folate drugs and CDK inhibitors. Importantly, we show that selective CDK4/6 inhibitors are pharmacological activators of SAMHD1 that act by inhibiting its inactivation by phosphorylation. Combinations of a CDK4/6 inhibitor with nucleoside or folate antimetabolites potently enhanced drug efficacy, resulting in highly synergic drug combinations (CI < 0.04). Mechanistic analyses reveal that cell cycle-controlled modulation of SAMHD1 function is the central process explaining changes in anticancer drug efficacy, therefore providing functional proof of the potential of CDK4/6 inhibitors as a new class of adjuvants to boost chemotherapeutic regimens. The evaluation of SAMHD1 expression in cancer tissues allowed for the identification of cancer types that would benefit from the pharmacological modulation of SAMHD1 function. In conclusion, these results indicate that the modulation of SAMHD1 function may represent a promising strategy for the improvement of current antimetabolite-based treatments.

## 1. Introduction

Sterile alpha motif and histidine-aspartic acid domain-containing protein 1 (SAMHD1) is a deoxynucleotide triphosphate hydrolase [1]. Through its dNTPase activity, SAMHD1 maintains the intracellular dNTP pool at levels adequate for DNA replication and repair but below a potentially mutagenic threshold [2]. Mutations in SAMHD1 were first identified as being causative of Aicardi–Goutieres syndrome (AGS), a severe autoimmune disease [3], but later it was proposed as a tumor suppressor gene, since mutations in SAMHD1 are associated with different types of cancer [4,5] and its expression has been found to be inhibited in several types of tumors [6,7]. More recently, SAMHD1 has been implicated in the protection against cancer and chronic inflammation by limiting the release of single-stranded DNA [8].

SAMHD1 also acts as a viral restriction factor, limiting the permissiveness of cells to diverse viruses (reviewed in [9]), including human immunodeficiency virus (HIV-1) [10,11]. SAMHD1 inhibits retroviral replication at the reverse transcription (RT) step by maintaining the intracellular concentration of dNTP below the threshold required for reverse transcription of the viral RNA genome into DNA [10,11]. SAMHD1 activity is counteracted by the HIV-2 accessory protein Vpx, a tool that has been extensively used to study SAMHD1 function [10,12]. In addition, SAMHD1 function is inhibited by phosphorylation, a process controlled by cyclin-dependent kinases (CDK) and therefore tightly regulated during cell cycle progression [13,14,15]. Different pharmacological agents have been shown to block SAMHD1 phosphorylation, inducing SAMHD1 function and viral restriction [16,17,18,19,20].

Nucleoside analogues are commonly used in the treatment of viral infections and cancer [21]. Following phosphorylation by intracellular kinases, these analogues are structurally similarly to endogenous dNTP. SAMHD1 has been shown to affect the efficacy of nucleoside analogs, either used as antiretrovirals [22,23,24,25,26] or as chemotherapeutic drugs [27,28,29]. Active SAMHD1 catalyzes the hydrolysis and inactivation of a number of different nucleoside analogues [30,31], including cytarabine (Cytosar-U^®^,Ara-C), a first line therapeutic agent for acute myelogenous leukaemia (AML) and SAMHD1 expression levels were negatively correlated with Ara-C treatment success in individuals with AML [27,29,32].

Here, we evaluated how SAMHD1 function modifies the efficacy of a wide range of nucleoside and non-nucleoside antimetabolites currently used to treat cancer. We show that SAMHD1 can either enhance or limit the efficacy of a number of chemotherapeutic drugs, allowing us to discern between those acting as enzyme substrates or competitors. Importantly, we identify anti-folate drugs as being affected by SAMHD1 expression and function. Moreover, pharmacological activation of SAMHD1 by highly selective CDK4/6 inhibitors significantly enhanced the efficacy of drugs acting as competitors. Thus, pharmacological modulation of SAMHD1 activity has the potential to improve antiviral and anti-cancer therapies and paves the way to the identification of malignancies that may be treated with new drug combinations.

## 2. Results

### 2.1. SAMHD1 Regulates Antiviral Efficacy of Antimetabolites in Primary Cells

Primary monocyte-derived macrophages (MDMs) are susceptible to HIV-1 infection, and its replication capacity is dependent on SAMHD1 expression. Additionally, M-CSF-induced differentiation initiates MDM proliferation accompanied by SAMHD1 phosphorylation. Thus, HIV-1 infection of MDM provides an excellent model in which to test the activity of antimetabolite drug efficacy.

Anti-HIV-1 activity of a panel of antimetabolite drugs used in cancer treatment was evaluated in MDMs in the presence or absence of SAMHD1, after transducing cells with HIV-2 Vpx (Figure 1A). As previously observed, Vpx-mediated degradation of SAMHD1 reduced the antiviral potency of the nucleoside reverse transcriptase inhibitor (NRTI) AZT compared to untreated macrophages, but did not change the activity of NVP, a non-nucleoside reverse transcriptase inhibitor (NNRTI) (Figure 1B). Conversely, degradation of SAMHD1 improved the anti-HIV-1 potency of AraC in MDM (Figure 1C).

Next, we tested the antiviral activity of a panel of antimetabolites currently used in cancer treatments in wild type or SAMHD1 depleted macrophages. All evaluated drugs inhibited HIV-1 replication, although with different potency (Table 1). SAMHD1 expression effectively modified the antiviral activity of all antimetabolites tested. However, and in contrast with previous reports, SAMHD1 degradation either enhanced (cladribine, clofarabine, and nelarabine) or decreased (capecitabine, floxuridine and fluorouracil) the potency of the nucleoside analogues tested (Figure 2A). Of note, SAMHD1 degradation dramatically impaired the efficacy of anti-folate inhibitors such as pemetrexed and methotrexate (Figure 2B). Calculation of 50% effective concentrations (EC_50_) of antimetabolites in macrophages expressing SAMHD1 or not showed over 30-fold and 100-fold increases in drugs showing enhanced or diminished potency in SAMHD1-depleted cells, respectively (Table 1). The enhanced or decreased efficacy of the compounds tested was not dependent on the nature of the specific nucleotide targeted, i.e., purine or pyrimidine, and was not limited to nucleos(t)ide analogues, as SAMHD1 also affected the efficacy of anti-folate drugs such as pemetrexed and methotrexate (Table 1).

### 2.2. SAMHD1 Is Required for Antiviral Activity of CDK4/6 Inhibitors

SAMHD1 is inactivated in proliferating cells by a mechanism that requires its phosphorylation [13,14,15]. SAMHD1 phosphorylation may be directly regulated by CDK1 or CDK2, whose activity is upstream controlled by CDK6 [14]. The antiviral activity of the highly selective CDK4/6 inhibitor palbociclib is dependent on SAMHD1 expression [14,19] (Figure 3A). Thus, the efficacy of two other specific CDK4/6 inhibitors, ribociclib and abemaciclib, was also evaluated in the presence or absence of SAMHD1. The three agents were tested at the concentration where palbociclib showed the highest efficacy in cell culture (1 μM, Figure 3A). As expected, the activity of all three CDK4/6 inhibitors was lost in the absence of SAMHD1, indicating that the efficacy of CDK4/6 inhibitors depends on SAMHD1 expression (Figure 3B). Interestingly, similar results were obtained when the multi-kinase inhibitor midostaurin was evaluated (Appendix A), suggesting that activity of multiple types of kinase inhibitors may be influenced by SAMHD1 expression.

To explore the cellular and molecular determinants of SAMHD1 requirement for kinase inhibitor function, SAMHD1 expression and phosphorylation was measured by Western blot. Both palbociclib and midostaurin blocked SAMHD1 phosphorylation, whereas SAMHD1 protein expression was not affected (Figure 3C, Appendix A). In addition, we observed a concomitant dephosphorylation and decreased expression of Rb, a substrate of CDK6, suggesting that palbociclib and midostaurin also affect CDK6-mediated CDK2 phosphorylation of SAMHD1 (Figure 3C, Appendix A).

### 2.3. Pharmacological Inhibition of CDK4/6 Enhances Antiviral Activity of Antimetabolites

CDK4/6 inhibitors activate SAMHD1 function through the inhibition of its phosphorylation. Thus, we evaluated the capacity of palbociclib to modify the activity of antimetabolites. Antiviral activity of pemetrexed and fluorouracil were evaluated alone or in combination with palbociclib in primary macrophages.

Pemetrexed inhibited HIV-1 replication in a dose-dependent manner, although with limited potency (EC_50_ = 0.1 µM, Figure 4A, black line). Combination of pemetrexed with increasing concentrations of palbociclib (EC_50_ = 0.12 µM) enhanced the antiviral potency of the antimetabolite (Figure 4A,B, left panels). The calculation of the combination index (CI) indicated strong synergy (CI ≤ 0.041 for palbociclib at 0.04 µM combined with different concentrations of pemetrexed, Table 2). Interestingly, pemetrexed and palbociclib activity, as well as the synergistic effect observed in drug combinations, were lost in the absence of SAMHD1, (Figure 4A,B, right panels). Furthermore, the combination of pemetrexed with the multi-kinase inhibitor midostaurin (EC_50_ = 0.62 µM) also showed a highly synergistic effect when SAMHD1 was expressed (Figure 4C,D, left panels, Table 2), an effect that was lost when SAMHD1 depleted cells (Figure 4C,D, right panels).

On the other hand, combination of the nucleoside analogue fluorouracil or the multikinase inhibitor midostaurin with palbociclib showed more limited effects, i.e., palbociclib partially enhanced the antiviral potency of fluorouracil or midostaurin in the presence of SAMHD1 (Appendix A). Combination index calculation indicated synergy at specific concentrations, although CI were 100-fold lower compared to palbociclib-pemetrexed drug interactions (Table 2) and antagonic or additive effects were also seen.

Overall, these results suggest that pharmacological activation of SAMHD1 can significantly enhance the efficacy of antimetabolites, through a mechanism that is dependent on SAMHD1 expression and regulation.

### 2.4. Cytotoxic Efficacy of Antimetabolites Is Enhanced by CDK4/6 Inhibitors

Anticancer drugs are specifically designed to inhibit cell growth, thus we evaluated cytotoxic efficacy of the antimetabolites pemetrexed and fluorouracil in combination with the CDK4/6 inhibitor palbociblib in the TZM-bl cell line and in two distinct breast cancer cell lines, MDA-MB-468 and T47D (Appendix A). As expected, all drugs tested resulted in decreased cell metabolic activity in all cell lines, reflecting the number of viable cells under defined conditions. The calculation of 50% cytotoxic concentrations (CC_50_) showed significant differences between them (Appendix A).

The combination of pemetrexed with palbociclib enhanced the cytotoxicity of the antimetabolite in all cell lines tested (Figure 5A–C). Importantly, the calculation of the combination index indicated a synergistic effect in all cases, with the cytotoxic evaluation being comparable to the results obtained when antiviral efficacy was measured (Table 3). The combination of fluorouracil with palbociclib enhanced fluorouracil potency in TZM-bl and T47D cells but not in MDA-MB-468 cell line (Figure 5A–C, Table 3). Interestingly, in MDA-MB-468 cells, although the expression of SAMHD1 was similar to other lines, Rb and pRb were not detected, either at the mRNA or protein level, (Appendix A), demonstrating the importance of cell cycle proteins which putatively may affect SAMHD1 function in determining palbociclib–antimetabolite drug combination efficacy.

### 2.5. Alternative Pathways of dNTP Metabolism Control Are Responsible for Drug Synergy

To further explore the mechanism underlying the synergistic effect observed when combining antimetabolites with the CDK4/6 inhibitor palbociclib, protein expression in primary macrophages treated with pemetrexed, fluorouracil and palbociclib, alone or in combination was evaluated (Figure 6A). As expected, palbociclib alone inhibited phosphorylation of pRb and SAMHD1, therefore activating its dNTP triphosphohydrolase function and subsequently reducing the intracellular dNTP pool. Interestingly, pemetrexed and fluorouracil treatment resulted in different effects, i.e., while fluorouracil acts similarly to palbociclib, pemetrexed did not decrease the phosphorylation of pRb and SAMHD1. Although pemetrexed activity is dependent on SAMHD1, its mechanisms of action do not directly affect SAMHD1 phosphorylation, providing evidence for the stronger synergy observed in the pemetrexed–palbociclib drug combination compared to fluorouracil–palbociblib.

Thus, antifolates such as pemetrexed inhibit the dNTP pool by a mechanism not directly affecting SAMHD1 phosphorylation and effectively synergize with palbociclib, which actually induces SAMHD1 activation. On the other hand, when two compounds directly affecting SAMHD1 phosphorylation (i.e., fluorouracil and palbociclib) are combined, the synergic effect is less potent.

### 2.6. SAMHD1 Is Expressed in Different Tumor Tissues

To further explore the potential value of modulating SAMHD1 function in cancer patients, we evaluated SAMHD1 expression in different tumor tissues by immunohistochemistry (IHC) in paraffin-embedded tissues. SAMHD1 was clearly detected in at least two cancer tissue types susceptible to being treated with antimetabolites—pancreatic adenocarcinoma and lung large cell carcinoma (Figure 7A,B). In both cases, SAMHD1 was significantly expressed in a high percentage of malignant cells.

In addition, IHC data of 17 different types of human tumors including 202 different samples from human protein atlas were also analyzed (www.proteinatlas.org, [33]). Although SAMHD1 was expressed in all types of tumors, the degree of expression was significantly variable, ranging from undetectable levels to high protein expression levels. Overall, 70% of all tumors expressed SAMHD1 to a certain extent, whereas its expression could not be detected in 30% of cases (Figure 7C). These results demonstrate that SAMHD1 is expressed in patient tumor samples but also suggest that modulation of SAMHD1 function might be feasible at least in a subgroup of cancer types.

## 3. Discussion

Nucleotide metabolism plays a central role in cell proliferation, transformation and tumor progression [34]. Therefore, inhibition of nucleotide synthesis has been commonly used in the treatment of cancer, infectious diseases and immune-mediated diseases [34]. By degrading cellular dNTPs, SAMHD1 plays an important role in the homeostatic balance of cellular dNTPs and, thus, it may be a modulator of clinical efficacy of nucleotide-based treatments. SAMHD1 has been shown to act as a resistance factor to nucleoside-based chemotherapies by hydrolyzing their active triphosphate metabolite, thereby reducing the response to the anticancer drugs [28,30].

Here, we show that SAMHD1 can either enhance or limit efficacy of a wide range of antimetabolites, including nucleoside analogues but also anti-folate drugs. Interestingly, antifolate drugs such as pemetrexed or methotrexate showed increased potency in SAMHD1-expressing cells, whereas nucleoside analogues efficacy was either limited or enhanced by SAMHD1. Attempts to predict the effect of SAMHD1 on drug efficacy based on the specific base targeted or chemical structure did not show any clear correlation, although purine nucleoside analogues may be more prone to gain activity in SAMHD1-depleted cells. Based on previous data showing that beyond endogenous dNTPs, triphosphorylated nucleoside analogues can be hydrolyzed by SAMHD1 [28,29,30,35] and considering that recombinant SAMHD1 exhibits Ara-CTPase activity in vitro [29], we hypothesized that compounds whose activity is enhanced in the absence of SAMHD1 are enzyme substrates. On the contrary, and as previously demonstrated for NRTI [23,24,25], compounds that lose activity in the absence of SAMHD1 would be competing with the intracellular dNTP pool, which is lower when SAMHD1 is active. Accordingly, anti-folates showed higher activity when SAMHD1 effectively limits the dNTP pool.

As previously reported, our data indicate that SAMHD1 decreases activity of drugs used against AML (AraC and clofarabine) and other types of hematological cancers like nelarabine for T-lymphoblastic lymphoma, cladribine used in the treatment of hairy cell leukemia and B-cell chronic lymphocytic leukemia, and approved for the treatment of relapsing-remitting multiple sclerosis. On the contrary, SAMHD1 enhances the activity of drugs used to treat solid tumors, such as floxuridine, most often used in the treatment of colorectal cancer, fluorouracil, used for colon cancer, esophageal cancer, stomach cancer, pancreatic cancer, breast cancer, and cervical cancer, and anti-folate drugs, such as methotrexate, used against a number of cancers but also to treat autoimmune diseases, such as psoriasis, rheumatoid arthritis, and Crohn’s disease, or pemetrexed, used to treat non-small cell lung cancer [36]. Thus, the data presented here expand the number of anticancer drugs known to be affected by SAMHD1 function and the number of diseases that could benefit from the development of pharmacological inhibitors or activators of SAMHD1 function.

It is known that SAMHD1 function is tightly linked to cell cycle control as its activity is regulated by CDK-dependent phosphorylation (reviewed in [9]). We and others have identified drugs that activate SAMHD1 function by impeding its phosphorylation [16,17,18,19,20]. Such drugs may be suitable for combination therapies including antimetabolites, especially when combined with anti-folate drugs, as synergistic effects were seen across all cell models tested. Detection of the phosphorylated form of SAMHD1 and expression of related cell cycle proteins allowed us to suggest a mode of action where anti-folate drugs would diminish the dNTP pool by affecting the synthesis of nucleotide precursors, whereas CDK inhibitors deplete dNTPs by activating SAMHD1 triphosphohydrolase activity [19] (Figure 6B). Although our data provide strong evidence of the key role of SAMHD1 in determining efficacy of CDK4/6 inhibitors, further work will have to determine the relative contribution of SAMHD1 in each type of tumor, by, for example, performing additional evaluations in SAMHD1 in vitro and/or in vivo knock-out models. From our data, we cannot rule out the possibility that synergy observed in breast cancer cell lines might be not entirely dependent on SAMHD1, but due to a combination of different factors that deserve further investigation.

Antimetabolites were the first class of cytotoxic drugs systematically tested in clinical trials that elicited complete clinical responses as monotherapies, albeit with inevitable relapse [37]. Combination chemotherapy still constitutes the current paradigm to achieve systemic disease control in clinical oncology. Thus, our results might represent the first step of a novel treatment strategy directed to the activation of SAMHD1 function. In this sense, the choice of highly selective CDK4/6 inhibitors as the first clinically effective CDK4/6 inhibitors developed [38,39] is of special relevance. Currently, inhibition of CDK4/6 in combination with endocrine therapies is the treatment option in hormone receptor-positive/HER2-negative advanced breast cancer. CDK4/6 inhibitors offer an effective and tolerable treatment that can be combined with other therapies and thus harbors therapeutic potential for multiple cancers [40,41]. Based on our data, CDK4/6 inhibitors could boost antimetabolite-based anticancer therapies, especially for drugs whose activity is enhanced by SAMHD1, which highlights the need for a priori testing drug efficacy depending on SAMHD1 function.

We have applied here a screening approach based in anti-HIV-1 activity in primary macrophages that either express SAMHD1 or do not. Our system has turned out to be a reliable and sensitive measure of SAMHD1 drug dependency for several reasons: (i) SAMHD1 expression is easily modulated, (ii) HIV-1 reverse transcription is a process highly sensitive to SAMHD1-mediated dNTP pool sizes changes and can be easily monitored, (iii) cell cycle initiation and progression is not deregulated and (iv) considers inter-individual differences by using primary cells from different donors. Indeed, cytotoxicity data obtained in different cell lines confirmed the results from the antiviral-based screening, except when expression of pRb was deregulated. RB is one of the key factors in cell cycle control that is phosphorylated by activation of CDK4 and CDK6 following cell cycle entry from GO to G1 [42], as is the case of SAMHD1.

## 4. Materials and Methods

### 4.1. Cells

Peripheral blood mononuclear cells (PBMC) were obtained from the blood of healthy donors using a Ficoll–Paque density gradient centrifugation and monocytes were purified using negative selection antibody cocktails (StemCell Technologies, Vancouver, BC, Canada) as described before [43]. Monocytes were cultured in complete culture medium: RPMI 1640 medium supplemented with 10% heat-inactivated fetal bovine serum (FBS; Thermo Fisher Scientific, Waltham, MA, USA), penicillin and streptomycin (Thermo Fisher Scientific, Waltham, MA, USA) and differentiated to monocyte derived macrophages (MDM) for 4 days in the presence of monocyte-colony stimulating factor (M-CSF, Peprotech) at 100 ng/mL. The protocol was approved by the Scientific Committee of Institut de Recerca de la Sida – IrsiCaixa and the Ethics Review Board of Hospital Germans Trias i Pujol. Buffy coats were purchased from the Catalan Banc de Sang i Teixits (http://www.bancsang.net/en/index.html). The buffy coats received were totally anonymous and untraceable and the only information given was whether or not they have been tested for disease. All donors provided informed consent at the time of blood extraction.

The human cell lines HEK293-T, TZM-bl (AIDS Reagent Program, National Institutes of Health, Bethesda, MD) MDA-MB-468 (ATCC^®^ HTB132™) and T47D (ATCC^®^ HTB-133™) (American Type Culture Collection, ATCC) were cultured in Dulbecco’s modified Eagle’s medium (DMEM; Gibco, Madrid, Spain) supplemented with 10% heat inactivated FCS, 100 U/mL penicillin, and 100 ug/mL streptomycin. SAMHD1 knock-out TZM cells were generated by CRISPR/Cas9 technique as described elsewhere (22).

### 4.2. Drugs

33-Azido-3-deoxythymidine (zidovudine, AZT), nevirapine (NVP), 1-beta-D-Arabinofuranosylcytosine (AraC), 9-beta-D-Arabinofuranosyl-2-fluoroadenine (fludarabine), 2-Amino-9-beta-D-arabinofuranosyl-6-methoxy-9H-purine (nelarabine), 2-Chloro-2′-deoxyadenosine (cladribine), 2-chloro-9-(2-Deoxy-2-fluoro-beta-D-arabinofuranosyl)adenine (clofarabine), 2′,2′-Difluorodeoxycytidine (gemcitabine), 5-Fluoro-2-desoxyuridine (floxuridine), 5-Fluoropyrimidine-2,4-dione (fluorouracil), pemetrexed and 4′-N-benzoylstaurosporine (midostaurin) were purchased from Sigma-Aldrich (Madrid, Spain). Palbociclib, ribociclib and abemaciclib were purchased from Selleckchem (Munich, Germany). 4-amino-10-methylfolic acid (methotrexate) was purchased from Eurodiagnosticos SL (Madrid, Spain).

### 4.3. Virus

Envelope-deficient HIV-1 NL4-3 clone encoding IRES-GFP (NL4-3-GFP) was pseudotyped with VSV-G by cotransfection of HEK293T cells using polyethylenimine (Polysciences, Warrington, PA, USA) as previously described [44]. For the production of viral-like particles carrying Vpx (VLP-Vpx), HEK293T cells were cotransfected with pSIV3+ and a VSV-G expressing plasmid. Three days after transfection, supernatants were harvested, filtered and stored at −80 °C. Viral stocks were concentrated using Lenti-X concentrator (Clontech, Mountain View, CA, USA). Viruses were titrated by infection of TZM cells followed by GFP quantification by flow cytometry.

### 4.4. Virus Infections

MDMs were pretreated with VLP-Vpx for 4h before infection or left with fresh media as a control. Cells were then infected with VSV-pseudotyped NL4-3-GFP and drugs were added at the time of infection. Viral replication was measured two days later by flow cytometry, as well as measurement of the viability of cells by gating live vs. dead cells. The anti-HIV activity of the different compounds was determined by infection of cells in the presence of different concentrations of the drug and 50% effective concentrations (EC_50_) were calculated, as previously described [23].

### 4.5. Western Blot

Cell extracts were prepared for Western blot protein analysis as described before [22]. Briefly, cells were rinsed in ice-cold phosphate-buffered saline (PBS) and extracts prepared in lysis buffer (50 mM Tris HCl pH 7.5, 1 mM EDTA, 1 mM EGTA, 1 mM Na3VO4, 10 mM Na β-glycerophosphate, 50 mM NaF, 5 mM Na Pyrophosphate, 270 mM sucrose and 1% Triton X-100) supplemented with protease inhibitor (Roche, Basel, Switzerland) and 1 mM phenylmethylsulfonyl fluoride. Lysates were subjected to SDS-PAGE and transferred to a PVDF membrane (ImmunolonP, Thermo, Waltham, MA, USA). The following antibodies were used for immunoblotting: anti-rabbit and anti-mouse horseradish peroxidase-conjugated secondary antibodies (1:5000; Pierce, Waltham, MA, USA; anti-human Hsp90 (BD Biosciences, Franklin Lakes, NJ, USA; 610418), anti-SAMHD1 (1:2500; ab67820, Abcam, Cambridge, UK) and anti-RRM2 (1:1000; ab115701, Abcam) and anti-phospho-CDK2 (Thr160; 2561), anti-CDK2 (2546), anti-CDK1 (77055) anti-Rb (9309) anti-phospho-Rb (Ser807/811, 9308) anti-E2F1 (3742), anti-phospho-SAMHD1 Thr592 (15038) (all 1:1000; Cell Signaling Technologies, Danvers, MA, USA).

### 4.6. Evaluation of Cytotoxicity

Cytotoxicity measurements in cell lines were based on the viability of cells that had been treated or not with various concentrations of the different anticancer compounds. Cells were treated at the indicated doses of the test compounds for 3 days and the number of viable cells was quantified by a tetrazolium-based colorimetric method (MTT method) as described elsewhere [23]. The MTT assay measured the metabolic activity of cells, resulting in a very sensitive procedure to evaluate cell viability and cell proliferation, including the effect of cytostatic agents that slow or stop cell growth.

### 4.7. Evaluation of Drug Combination

Drug combinations were evaluated using the combination index (CI)-isobologram equation, a method widely used in pharmacology to study drug interactions. Relative values of drug activity were used to calculate CI as implemented in the Compusyn software (Combosyn Inc., Paramus, NJ, USA) [45]. In brief, combination experiments were performed by using serial dilutions of each drug alone or a mixture of the two drugs evaluated, as recommended by the Chou–Talay method using a non-constant ratio combination [46]. CI was calculated for all combinations and those combinations, including concentrations of SAMHD1-activating drugs around calculated IC_50_, were considered for quantification of drug combination effect. Drug combinations with CI < 1 were considered synergic [46].

### 4.8. Immunohistochemistry

Tissue sections from lung and pancreas tumor tissues (T2235188-1 and T2235152, respectively, Amsbio UK), were used to evaluate SAMHD1 expression in tumor tissues ex vivo. All immunohistochemical analyses were performed in the Histopatology core facility at Germans Trias i Pujol Research Institute. A polyclonal rabbit anti-SAMHD1 antibody (cat. no. 12586-1-AP, Proteintech, Rosemont, IL, USA) and an automated detection system were utilized. The specificity of the polyclonal antibody was previously tested by Western blot analysis in cell lines and by immunohistochemistry using paraffin-embedded normal tissue. Images were obtained in a Zeiss Axioskop 2 microscope using ZEN blue 2011 software.

### 4.9. Statistical Methods

Data were analyzed with the PRISM statistical package. If not stated otherwise, all data were normally distributed and expressed as mean ± SD. p-values were calculated using an unpaired, two-tailed, t-student test.

## 5. Conclusions

Our results confirm and expand the number of drugs whose efficacy might be affected by SAMHD1 function, demonstrating the importance of SAMHD1 for targeted health therapies including nucleoside analogues such as treatments for cancer and viral infections. Moreover, we show that pharmacological modulation of SAMHD1 can significantly enhance current antimetabolite-based anticancer therapies. Nucleobase and nucleoside analogues are an important class of anti-cancer and antiviral drugs. Thus, understanding the clinical and molecular determinants of drug efficacy is paramount to improve the efficacy of anticancer treatment. Based on our findings, the development of robust SAMHD1 inhibitors and activators that can potentiate antimetabolite therapeutic regimens should become a priority.

## Figures and Tables

**Figure 1 cancers-12-00713-f001:**
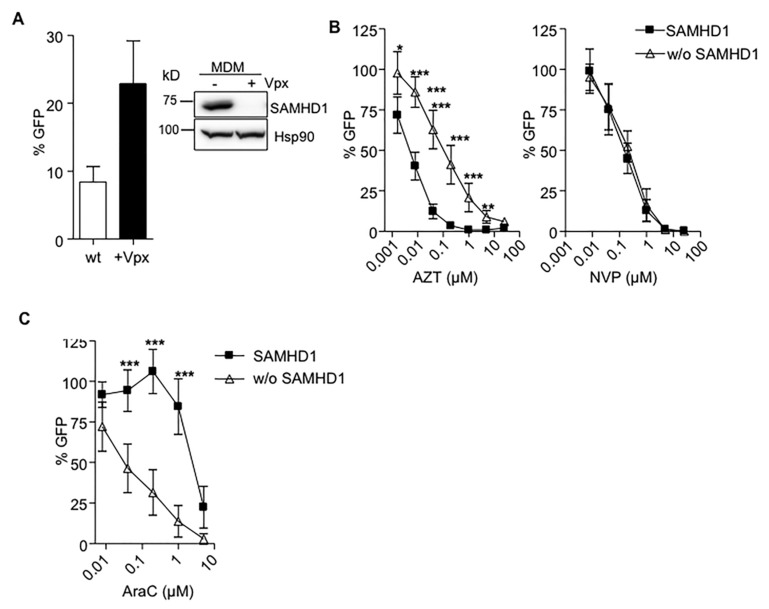
(**A**) Degradation of sterile alpha motif and histidine-aspartic acid domain-containing protein 1 (SAMHD1) by HIV-2 Vpx enhances HIV-1 replication in monocyte-derived macrophages (MDMs). MDMs previously treated or not with HIV-2 Vpx were infected with a VSV-pseudotyped HIV-1 GFP virus and replication assessed 2 days later by measuring GFP expression. A 5-fold change in HIV-1 replication was observed after Vpx-mediated SAMHD1 degradation. Mean ±SD of ten independent donors performed in duplicate is shown. A representative western blot showing degradation of SAMHD1 expression in MDMs after Vpx treatment is also shown. (**B**) Decreased sensitivity of AZT after Vpx-mediated SAMHD1 degradation in MDMs. Dose response of the NRTI AZT and NNRTIs NVP, in wild-type (■) or SAMHD1-depleted (Δ) MDMs. Inhibition of HIV infection was measured as the percentage of GFP+ cells relative to the no drug condition. Mean ±SD of at least ten independent donors performed in duplicate is shown. (**C**) SAMHD1 modifies antiviral activity of AraC. Dose response of the AraC in wild-type (■) or SAMHD1-depleted (Δ) MDM. Inhibition of HIV infection was measured as the percentage of GFP+ cells relative to the no drug condition. Mean ±SD of at least three independent donors performed in duplicate is shown. Mean ±SD of at least three independent experiments performed in triplicate is shown. * *p* < 0.05; ** *p* < 0.005; *** *p* < 0.0005.

**Figure 2 cancers-12-00713-f002:**
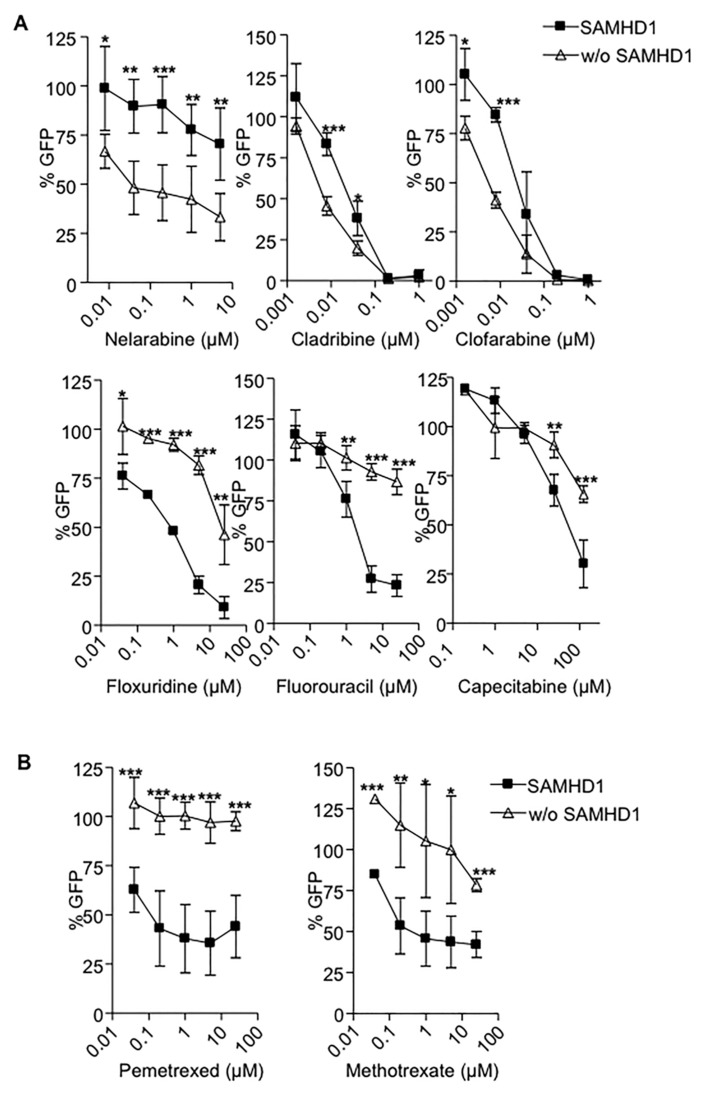
SAMHD1 modifies antiviral and activity of antimetabolites. Dose response of the nucleoside analogues (**A**) or anti-folate drugs (**B**) currently used as anti-cancer treatments in wild-type (■) or SAMHD1-depleted (Δ) MDMs. Inhibition of HIV infection was measured as the percentage of GFP+ cells relative to the no drug condition. Mean ±SD of at least three independent donors performed in duplicate is shown. * *p* < 0.05; ** *p* < 0.005; *** *p* < 0.0005.

**Figure 3 cancers-12-00713-f003:**
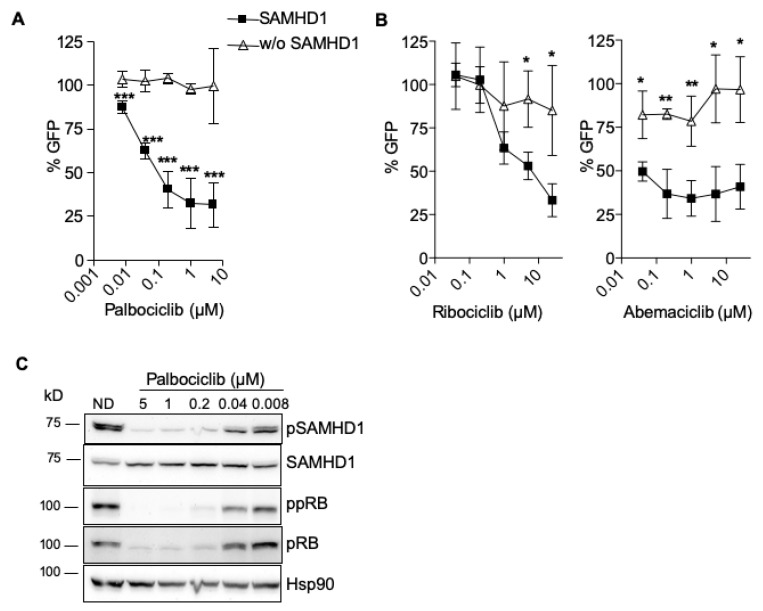
Antiviral efficacy of CDK4/6 inhibitors depends on SAMHD1 expression. (**A**) Dose response of the CDK4/6 inhibitor Palbociclib, in wild-type (■) or SAMHD1-depleted (Δ) MDM. Inhibition of HIV infection was measured as the percentage of GFP+ cells relative to the no drug condition. Mean ±SD of at least ten independent donors performed in duplicate is shown. (**B**) CDK4/6 inhibitors lose antiviral activity in SAMHD1-depleted macrophages. As in (**A**), dose response of two other CDK4/6 inhibitors, ribociclib (left panel) and abemaciclib (right panel), in wild-type (■) or SAMHD1-depleted (Δ) MDMs. Mean ±SD of two independent donors performed in duplicate is shown. (**C**) Palbociclib blocks SAMHD1 inactivation by phosphorylation. Western blot analysis of lysates of untreated MDMs (no drug, ND) or macrophages treated with palbociclib at the indicated doses. Membranes were blotted with an anti phospho-SAMHD1 antibody, total SAMHD1, anti phosho-pRB and total pRB. Hsp90 antibody was used as control. A representative donor is shown. * *p* < 0.05; ** *p* < 0.005; *** *p* < 0.0005.

**Figure 4 cancers-12-00713-f004:**
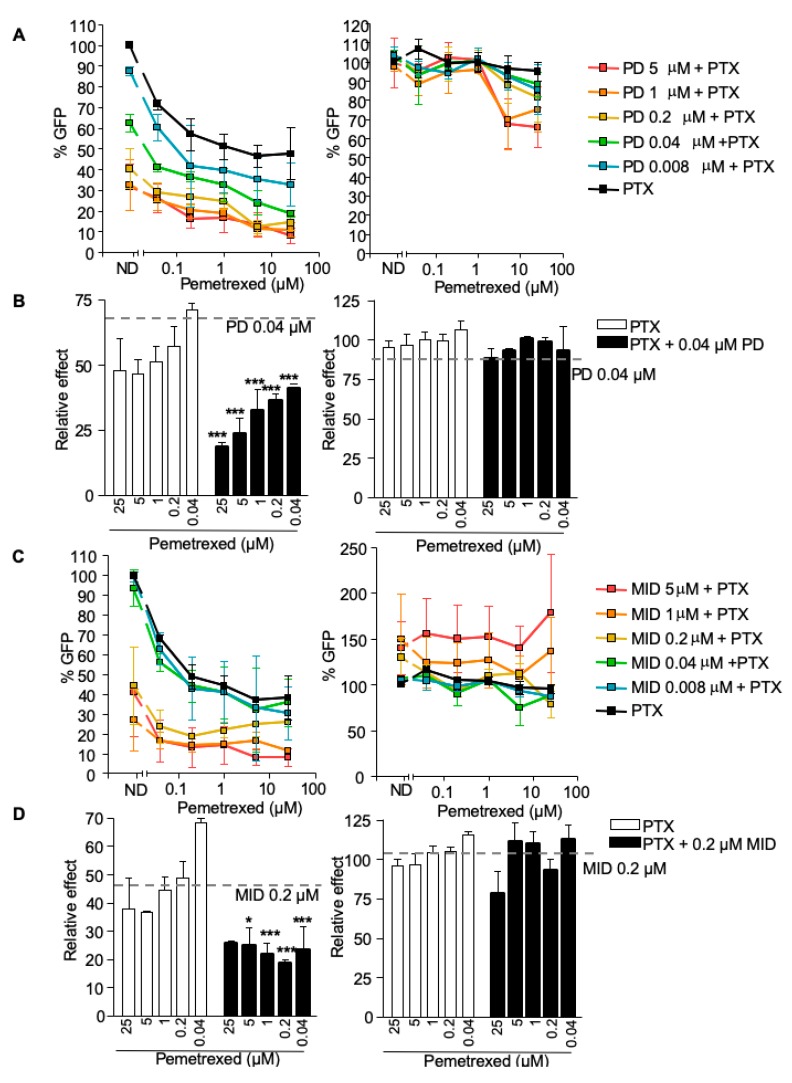
Pharmacological activation of SAMHD1 enhances antiviral activity of antimetabolites. (**A**) Relative effect of the combination of palbociclib-pemetrexed measured as antiviral activity. Inhibition of HIV infection with increasing doses of palbociclib and pemetrexed was measured. Percentage of GFP+ cells relative to the no drug condition is shown in presence (left panel) or absence (right panel) of SAMHD1. Mean ±SD of at least three independent donors performed in duplicate is shown. (**B**) As in (**A**), relative effect of pemetrexed alone (white bars) or in combination with a fixed dose of palbociclib 0.04 µM (black bars), in the presence (left panel) or absence (right panel) of SAMHD1. Mean ±SD of at least three independent donors performed in duplicate is shown. (**C**) Relative effect of the combination of midostaurin-pemetrexed measured as antiviral activity. Inhibition of HIV infection with increasing doses of midostaurin and pemetrexed was measured. Percentage of GFP+ cells relative to the no drug condition is shown in presence (left panel) or absence (right panel) of SAMHD1. Mean ±SD of at least three independent donors performed in duplicate is shown. (**D**) As in (**C**), relative effect of pemetrexed (PTX) alone (white bars) or in combination with a fixed dose of midostaurin 0.2 µM (black bars), in the presence (left panel) or absence (right panel) of SAMHD1. Mean ±SD of at least three independent donors performed in duplicate is shown. PD, palbociclib; PTX, pemetrexed; MID, midostaurin. * *p* < 0.05; ** *p* < 0.005; *** *p* < 0.0005.

**Figure 5 cancers-12-00713-f005:**
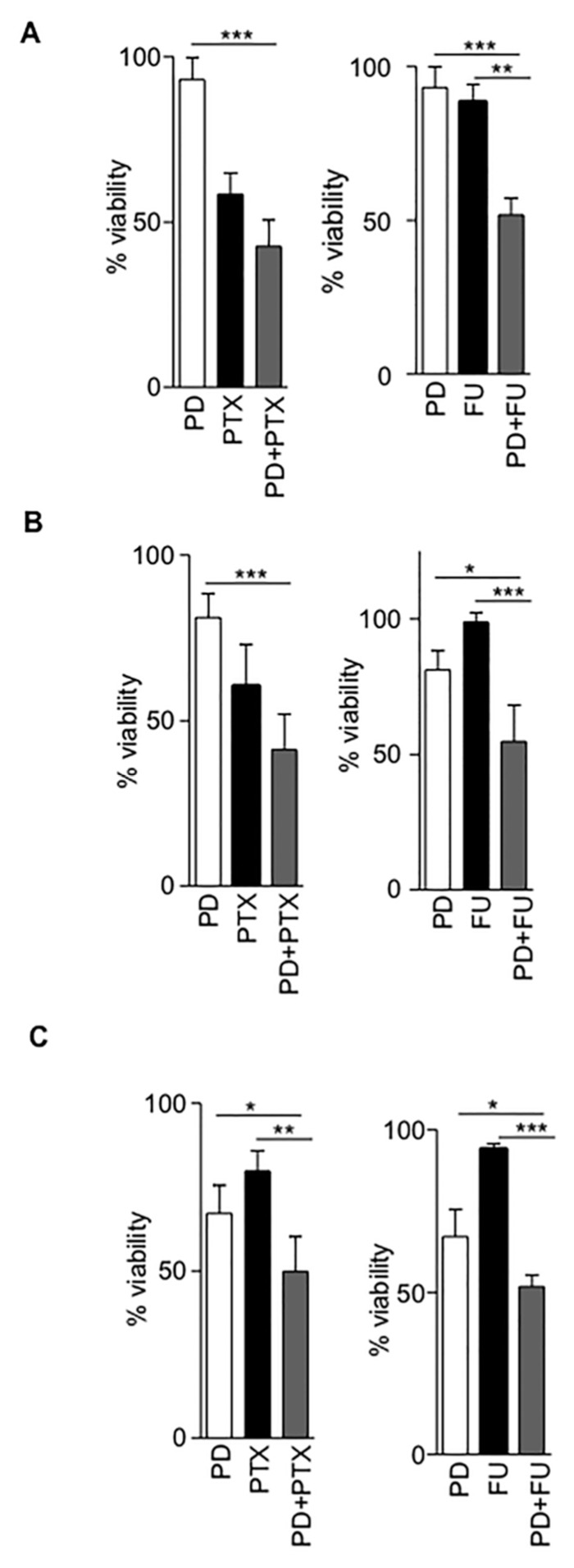
Pharmacological activation of SAMHD1 enhances cytotoxicity of antimetabolites. (**A**–**C**) Effect on cell viability of palbociclib–pemetrexed combination in TZM-bl, T47D and MDA-MB-468 cell lines, respectively. Left panels, cytotoxic activity of palbociclib alone (5 µM, white bars), pemetrexed alone (black bars at 0.2, 1 or 0.04 µM for TZM-bl, T47D and MDA-MB-468 respectively) or the combination of both drugs at the same concentration (grey bars). Right panels, cytotoxic activity of palbociclib alone (5 µM, white bars), fluorouracil alone (5 µM, black bars) or the combination of both drugs at the same concentration (grey bars). Drug concentrations were chosen depending calculated CC_50_ under specific experimental conditions (Appendix A). * *p* < 0.05; ** *p* < 0.005; *** *p* < 0.0005. PD, palbociclib; PTX, pemetrexed; FU, 5-fluorouracil.

**Figure 6 cancers-12-00713-f006:**
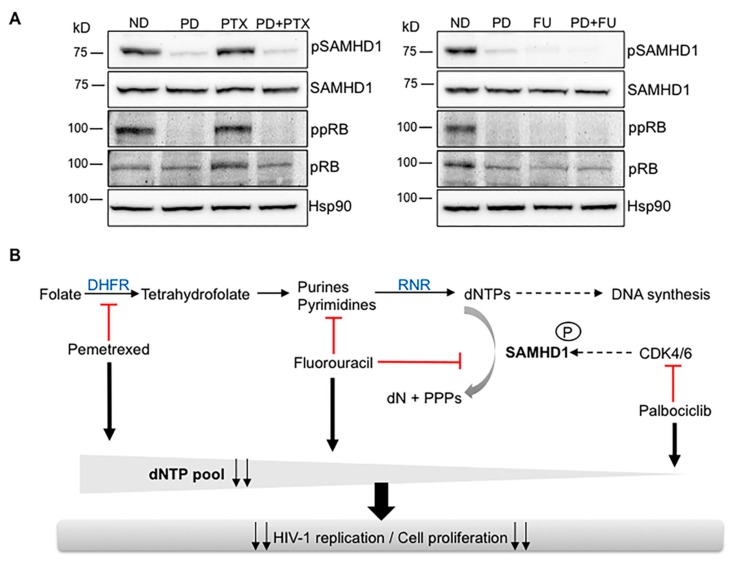
Regulation of dNTP pool is responsible for drug synergy. (**A**) Protein expression in MDMs treated with palbociclib (PD) at 1 µM, pemetrexed (PTX) and fluorouracil (FU), both at 5µM and the corresponding drug combinations PD+PTX and PD+FU. Hsp90 was used as a loading control. A representative blot is shown. (**B**) Proposed model of drug interactions. Antimetabolites affecting dNTP synthesis such as pemetrexed inhibit dNTP pool by a mechanism not directly affecting SAMHD1 activation and thus synergy with anticancer drugs affecting SAMHD1 phosphorylation as palbociclib is higher compared to compounds also targeting SAMHD1 function (i.e., fluorouracil) or exclusively affecting SAMHD1 (i.e., midostaurin). As a consequence, the antiviral and cytotoxic efficacy of antimetabolites is significantly enhanced when used in combination in vitro.

**Figure 7 cancers-12-00713-f007:**
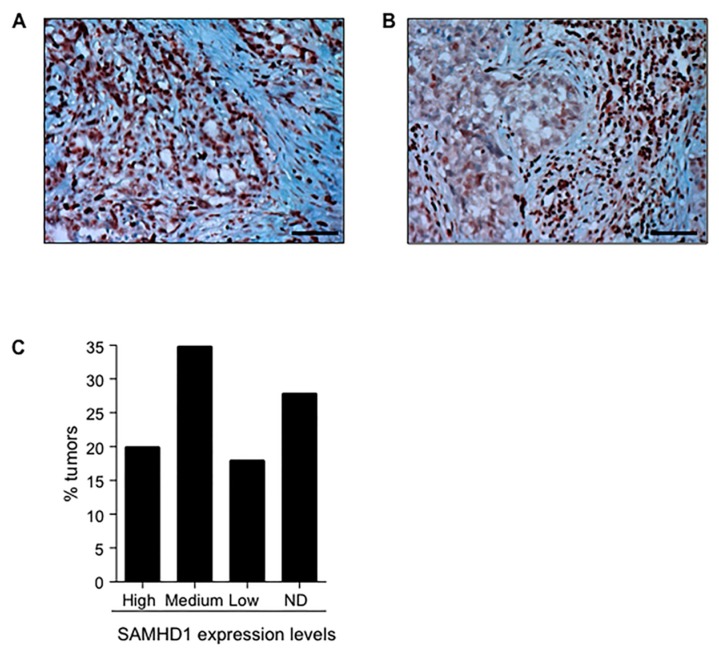
Expression of SAMHD1 protein in tumor samples from cancer patients. (**A**,**B**) IHC staining of SAMHD1 in pancreas (**A**) and lung (**B**) tumor samples. Morphology of tumor cells is shown by routine hematoxylin stain of paraffin embedded tumor sections. SAMHD1 is stained in brown. Original magnification ×200. Scale bar, 20 µm. (**C**) Percentage of tumors expressing SAMHD1, depending on its relative expression in IHC as classified in Human Proteome Atlas (www.proteinatlas.org, IHC data of SAMHD1 expression from 17 different tumor types were retrieved and classified according to protein expression levels. ND, not detected.

**Table 1 cancers-12-00713-t001:** Anti-HIV-1 activity of antimetabolites (AraC, fludarabine, nelarabine, cladribine, clofarabine, gemcitabine, floxuridine, fluorouracil, pemetrexed and methotrexate) and antiretrovirals (AZT and NVP) in macrophages expressing or not SAMHD1. The values of EC_50_ were calculated in primary MDMs untreated or transduced with HIV-2 Vpx that led to SAMHD1 degradation.

Drugs	Drug Type (Base Targeted)	EC_50_ (µM)		FC	
		SAMHD1 (+)	SAMHD1 (-)	-/+	+/-
AZT	NRTI (dT)	0.006	0.11	18	-
NVP	NNRTI (none)	0.88	0.95	1	1
AraC	Pyrimidine (dC)	3.24	0.11	-	30
Nelarabine	Purine (dG)	13.96	1.83	-	8
Cladribine	Purine (dA)	0.029	0.007	-	4
Clofarabine	Purine (dG)	0.034	0.006	-	6
Floxuridine	Pyrimidine (dU)	0.73	20.28	28	-
Fluorouracil	Pyrimidine (dU)	2.40	>25	>10	-
Pemetrexed	Anti-folate	0.25	>25	>100	-
Methotrexate	Anti-folate	0.42	79.24	190	-

EC_50_; Effective concentration required to block HIV-1 replication by 50%, FC; fold change or ratio of the EC_50_ without SAMHD1 and the EC_50_ with SAMHD1 (-/+), or inversely (+/-).

**Table 2 cancers-12-00713-t002:** Combination index values for pemetrexed and fluorouracil combinations with palbociclib and midostaurin. Macrophages were treated with the different at indicated concentrations in combination with 0.04 µM of palbociclib or 0.2 µM of midostaurin and infected with HIV-1 for 48 h. Drug efficacy was analyzed by measuring inhibition of HIV-1 replication by flow cytometry.

Drug Combination		Combination Index (CI)	Effect
Pemetrexed	25	0.0049	Synergy
+	5	0.0097	Synergy
Palbociclib 0.04 µM	1	0.0285	Synergy
	0.2	0.0415	Synergy
	0.04	0.0673	Synergy
Pemetrexed	25	0.079	Synergy
+	5	0.069	Synergy
Midostaurin 0.2 µM	1	0.056	Synergy
	0.2	0.045	Synergy
	0.04	0.064	Synergy
Fluorouracil	5	1.871	Antagonism
+	1	0.572	Synergy
Palbociclib 0.04 µM	0.2	0.658	Synergy
	0.04	1.818	Antagonism
	0.008	2.967	Antagonism
Midostaurin	5	2.074	Antagonism
+	1	0.427	Synergy
Palbociclib 0.04 µM	0.2	0.223	Synergy
	0.04	0.419	Synergy
	0.008	0.324	Synergy

CI values were calculated using the mean values of three different experiments. Values were calculated using CompuSyn software. CI < 1, synergy; CI > 1, antagonism; CI = 1, additive.

**Table 3 cancers-12-00713-t003:** Combination index values for pemetrexed and fluorouracil combinations with palbociclib. TZM-bl, MDA-MB-468 and T47D cells were treated with indicated concentrations of pemetrexed or fluorouracil in combination with 5 µM of palbociclib. Cytotoxic effect of the drugs was tested by MTT assay.

Cell Type	Drug Combination		Combination Index (CI)	Effect
**TZM-bl**	Pemetrexed	1	0.806	Synergy
	+	0.2	0.689	Synergy
	Palbociclib 5 µM	0.04	0.71	Synergy
		0.008	0.72	Synergy
	Fluorouracil	5	0.726	Synergy
	+	1	0.766	Synergy
	Palbociclib 5 µM	0.2	0.764	Synergy
		0.04	0.735	Synergy
**T47D**	Pemetrexed	5	0.783	Synergy
	+	1	0.707	Synergy
	Palbociclib 5 µM	0.2	0.779	Synergy
		0.04	0.874	Synergy
	Fluorouracil	25	0.848	Synergy
	+	5	0.804	Synergy
	Palbociclib 5 µM	1	0.966	Additive
		0.2	1.01	Additive
**MDA-MB-468**	Pemetrexed	1	0.886	Synergy
	+	0.2	0.886	Synergy
	Palbociclib 5 µM	0.04	0.921	Synergy
		0.008	1.01	Additive
	Fluorouracil	25	2.844	Antagonism
	+	5	1.745	Antagonism
	Palbociclib 5 µM	1	1.356	Antagonism
		0.2	1.125	Antagonism

CI values were calculated using the mean values of three different experiments. Values were calculated using CompuSyn software. CI < 1, synergy; CI > 1, antagonism; CI = 1, additive.

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
