# Peer review of "Pharmacological Modulation of SAMHD1 Activity by CDK4/6 Inhibitors Improves Anticancer Therapy"

_cancers, 2020, doi:10.3390/cancers12030713_

Round 1

Reviewer 1 Report

This study is hypothesis driven and provides mechanistic insight into the role of SAMHD1 in the regulation of hiv replication and anti-metabolite-based therapy efficacy. The study provides rationale for the use of CDK4/6 inhibitors to improve the efficacy of anti-metabolite and anti-viral treatments. Several places in the manuscript need further clarification:

  1. Concentration of palbociclib is different through out the paper. Fig 3B, 1uM is used however, in 3C is observed that phosphorylation of SAMHD1 is lost at 0.2uM. Not clear why higher concentration is shown in 3B
  2. Figure 5, 5uM of palbociclib is used. That is extremely high for T47D cells where the IC50 is in the 100nM range. Combination index shown in table 3 is also done with 5uM palbociclib. Does palbociclib have an effect used at lower concentrations?
  3. Figure 5, not clear that the results reflect cell viability. Palbociclib has a known cytostatic effect as opposed to a cytotoxic effect. The authors should revisit the results and confirm that cell death is observed. A more appropriate read out will be apoptosis assay or an equivalent assay that measures cell death and not MTT assay.
  4. Figure 5 and table 3, no results are shown in the absence of SAMHD1 (via siRNA). this is a critical control missing.

Author Response

We thank the reviewer for useful comments. Below we provide a point-by-point response to reviewer's comments.

This study is hypothesis driven and provides mechanistic insight into the role of SAMHD1 in the regulation of hiv replication and anti-metabolite-based therapy efficacy. The study provides rationale for the use of CDK4/6 inhibitors to improve the efficacy of anti-metabolite and anti-viral treatments. Several places in the manuscript need further clarification.

1.Concentration of palbociclib is different through out the paper. Fig 3B, 1uM is used however, in 3C is observed that phosphorylation of SAMHD1 is lost at 0.2uM. Not clear why higher concentration is shown in 3B

We agree with the reviewer that SAMHD1 phosphorylation is lost at 0.2 uM of palbociclib (Fig 3C). However, the highest efficacy of palbociclib according to cell culture data is observed at 1 uM, observing no significant increase in efficacy at higher concentrations (Fig 3A). Thus, in Fig 3B we choose the concentration that showed maximum efficacy (1 uM) to evaluate and compare the relative efficacy of all CDK4/6 inhibitors. We have now clarified this issue in results section (page 5, line 131-132).

2. Figure 5, 5uM of palbociclib is used. That is extremely high for T47D cells where the IC50 is in the 100nM range. Combination index shown in table 3 is also done with 5uM palbociclib. Does palbociclib have an effect used at lower concentrations?

As noted by the reviewer, reported palbociclib IC50 in T47D cells substantially differ between reports, ranging from 0,127 uM (PMID: 19874578 as an example) to 30 uM (Sanger institute data www.cancerrxgene.org), putatively indicating experimental differences such as detection method and time of exposure between others. However, in our experimental conditions, i. e., 50.000 cells/well, 3 days incubation and MTT method as a readout, palbociblib IC50 was 9,5 uM (Supplementary Table 1). Thus, evaluation of synergy in cell culture was perfomed at 5 uM, which was below our own calculated IC50.  As above, we have clarified this issue in page 9, line 228-229.

3. Figure 5, not clear that the results reflect cell viability. Palbociclib has a known cytostatic effect as opposed to a cytotoxic effect. The authors should revisit the results and confirm that cell death is observed. A more appropriate read out will be apoptosis assay or an equivalent assay that measures cell death and not MTT assay.

We thank the reviewer for the comment. MTT assay measures metabolic activity of the cells, resulting in a very sensitive and widely used procedure for measuring cell viability and cell proliferation, including the effect of cytostatic agents that slow or stop cell growth, as palbociclib (PMID: 31330844, 30605443, 30551429, as examples). Although we are aware that MTT assay does not measure apoptosis, the aim of our work was not to evaluate apoptotic pathways induced after palbociclib treatment rather to measure cell viability of palbociclib in combination with other anticancer agents. A sentence has been added in the results (page 8, line 209) and material and methods (page 14, line 407-408) sections to avoid misunderstandings.

4. Figure 5 and table 3, no results are shown in the absence of SAMHD1 (via siRNA). this is a critical control missing.

We evaluated efficacy of the tested drugs in the presence or absence of SAMHD1 in our macrophage model (Fig 1-4 and Tables 1-2), using HIV-2 Vpx viral like particles (VLP-Vpx) that are able to efficiently and stably degrade SAMHD1. Unfortunately, VLP-Vpx are not able to efficiently transduce the cell lines tested in Fig 5 and thus, the generation of knock-out clones by CRISPR/Cas9 is the only alternative. Although of interest, we believe that data in Fig 1-4 provide strong evidences of SAMHD1 key role in determining CDK4/6 inhibitor efficacy. We have included a sentence discussing this issue in the discussion section (page 13, lines 327-330).  

Reviewer 2 Report

The  manuscript, "Pharmacological modulation of SAMHD1 activity by CDK4/6 inhibitors improves anticancer theapy," by Castellvi et al., investigates how modulation of SAMHD1 levels and function affect the activity of various chemotherapeutic agents. Authors apply a screening approach to measure SAMHD1 drug dependency that is based in anti-HIV-1 activity in primary macrophages expressing or not expressing SAMHD1; this approach is based in part on the fact that HIV-1 reverse transcription is highly sensitive to SAMHD1-mediated dNTP pool size changes and this process can be easily measured. 

This is an informative study that adds to existing reports suggesting potential of SAMHD1 as a therapeutic target. I have the following suggestions/concerns: 

Figure 2B: Error bars for the methotrexate experiment are unusually large in comparison to the other experiments shown. 

Table 1 (page 5): For AZT, the EC50s for SAMHD1 (+) and SAMHD1 (-) are almost the same (0.1 versus 0.11), but the fold change reported is 18. This drug was shown in Figure 1 to be less effective in the presence of SAMHD1. For other drugs shown in the table, the fold changes match the EC50s reported. The EC50s reported for AZT are presumably a mistake. 

Figure 3 (page 6): Is the change in efficiency of ribociclib significant? Error bar for w/o SAMHD1 is big and overlaps with the + SAMHD1 treatment. 

Figure 4. What is PD? Is it palbociclib? The abbreviation needs to be defined. 

Tables 2 and 3: Perhaps authors could give some background about CompuSyn and how it works, compared to Calcusyn. For instance, how were 5-fold differences in the concentrations of one drug chosen for synergy experiments? How was the fixed concentration of the other drug chosen? The combination indices for certain drug combinations look strongly synergistic (<0.1). How might these values, generated by holding one concentration steady and combining with multiple concentrations of a second drug, compare with drug A and drug B tested in combination across multiple concentrations (according to the method of Chou and Talalay)?

CompuSyn (Chou TC; Martin N CompuSyn for drug combinations: PC Software and User's Guide: a computer program for quantitation of synergism and antagonism in drug combinations, and the determination of IC50 and ED50 and LD50 values, Combo-Syn: Paramus, NJ, USA, 2005; Chou, T-C AM. J. Cancer Res. 1:925-954, 2011) is a program that seems to be quite similar to Calcusyn software (based on Chou and Talalay, Adv Enzyme Regul. 1984;22:27-55). In the Compusyn product literature, it is suggested to use 5-6 drug concentrations of each drug being tested alone and in combination for synergy, centered around the IC50 for a constant ratio: 

"Two-fold serial dilutions are usually performed for in vitro experiment for each drug alone and their mixture to create 5-6 concentrations, with their Dm values (IC50 values) located in about the middle of the concentration ranges. The diluted mixture (e.g., in [IC50)1/(IC50)2] ratio is always at a constant ratio when it is diluted. The non-constant ratio combination can also be used for CI calculation. But it is less efficient in analysis or in computation due to changing ratios, which does not allow a simulation."

In Castellvi et al., authors take one fixed concentration of drug A and combine it with different concentrations of drug B (that are 5-fold apart) and report individual combination indices for each of the combinations and define them as "synergy" or "antagonism". Is the approach the authors are using classified as a "non-constant ratio combination"? Concentration ranges used for one of the two drugs being investigated for synergy are not two-fold serial dilutions, and authors do not use 5-6 concentrations of both drugs (only one), and it is unclear how close to the IC50 the middle of the concentration range of the one drug comes. It is also unclear how reliable the combination indices are compared to combination indices generated by use of two drugs combined in fixed ratios across 5-6 concentrations. 

Figure 5. Where is Figure 5D-F (page 9, line 215)? D-F is missing from Figure 5 and the Figure 5 legend.  

Author Response

The manuscript, "Pharmacological modulation of SAMHD1 activity by CDK4/6 inhibitors improves anticancer theapy," by Castellvi et al., investigates how modulation of SAMHD1 levels and function affect the activity of various chemotherapeutic agents. Authors apply a screening approach to measure SAMHD1 drug dependency that is based in anti-HIV-1 activity in primary macrophages expressing or not expressing SAMHD1; this approach is based in part on the fact that HIV-1 reverse transcription is highly sensitive to SAMHD1-mediated dNTP pool size changes and this process can be easily measured. This is an informative study that adds to existing reports suggesting potential of SAMHD1 as a therapeutic target. I have the following suggestions/concerns: 

1. Figure 2B: Error bars for the methotrexate experiment are unusually large in comparison to the other experiments shown. 

Although error bars for methotrexate are larger than those observed with the other antifolate agent tested, pemetrexed, differences between conditions (with and without SAMHD1) are statistically significant, indicating that SAMHD1 is similarly affecting efficacy of both drugs.

2. Table 1 (page 5): For AZT, the EC50s for SAMHD1 (+) and SAMHD1 (-) are almost the same (0.1 versus 0.11), but the fold change reported is 18. This drug was shown in Figure 1 to be less effective in the presence of SAMHD1. For other drugs shown in the table, the fold changes match the EC50s reported. The EC50s reported for AZT are presumably a mistake. 

We thank the reviewer for detecting the mistake. AZT data in table 1 has now been corrected.

3. Figure 3 (page 6): Is the change in efficiency of ribociclib significant? Error bar for w/o SAMHD1 is big and overlaps with the + SAMHD1 treatment. 

As correctly noted by the reviewer, ribociclib is not significant. This is in accordance with previous data showing that ribociclib is slightly less effective than palbociclib (PMID: 30443290). For clarity, we have added statistics to Fig 3B.

4. Figure 4. What is PD? Is it palbociclib? The abbreviation needs to be defined. 

PD is palbociclib. Abbreviations have now been included in all figure legends were necessary.

5. Tables 2 and 3: Perhaps authors could give some background about CompuSyn and how it works, compared to Calcusyn. For instance, how were 5-fold differences in the concentrations of one drug chosen for synergy experiments? How was the fixed concentration of the other drug chosen? The combination indices for certain drug combinations look strongly synergistic (<0.1). How might these values, generated by holding one concentration steady and combining with multiple concentrations of a second drug, compare with drug A and drug B tested in combination across multiple concentrations (according to the method of Chou and Talalay)? CompuSyn (Chou TC; Martin N CompuSyn for drug combinations: PC Software and User's Guide: a computer program for quantitation of synergism and antagonism in drug combinations, and the determination of IC50 and ED50 and LD50 values, Combo-Syn: Paramus, NJ, USA, 2005; Chou, T-C AM. J. Cancer Res. 1:925-954, 2011) is a program that seems to be quite similar to Calcusyn software (based on Chou and Talalay, Adv Enzyme Regul. 1984;22:27-55). In the Compusyn product literature, it is suggested to use 5-6 drug concentrations of each drug being tested alone and in combination for synergy, centered around the IC50 for a constant ratio: "Two-fold serial dilutions are usually performed for in vitro experiment for each drug alone and their mixture to create 5-6 concentrations, with their Dm values (IC50 values) located in about the middle of the concentration ranges. The diluted mixture (e.g., in [IC50)1/(IC50)2] ratio is always at a constant ratio when it is diluted. The non-constant ratio combination can also be used for CI calculation. But it is less efficient in analysis or in computation due to changing ratios, which does not allow a simulation." In Castellvi et al., authors take one fixed concentration of drug A and combine it with different concentrations of drug B (that are 5-fold apart) and report individual combination indices for each of the combinations and define them as "synergy" or "antagonism". Is the approach the authors are using classified as a "non-constant ratio combination"? Concentration ranges used for one of the two drugs being investigated for synergy are not two-fold serial dilutions, and authors do not use 5-6 concentrations of both drugs (only one), and it is unclear how close to the IC50 the middle of the concentration range of the one drug comes. It is also unclear how reliable the combination indices are compared to combination indices generated by use of two drugs combined in fixed ratios across 5-6 concentrations. 

We thank the reviewer for the comments on combination index calculations. As correctly noted by the reviewer, the only difference with calcusyn is that compusyn is the open-access not-for–profit purpose software based in the Chou Talay method for the quantification of drug combination effects, being both softwares essentially the same. For the design and analysis of drug combination experiments, we have followed recommendations stated in the compusyn product literature, based in the Chou Talay work. Indeed, all experiments have been performed with different concentrations of both drugs (see Figure 4 and supplementary Figure 2), and calculation of the combination index was performed in all cases. However, we have only included in the tables the combinations that were more informative, i. e., the concentration of palbociclib or midostaurin just below the mean IC50 of all individual experimental determinations.  To clarify these issues, IC50s of all drugs are now stated in the text were appropriate and a more detailed description on the procedures used (compusyn software and in vitro experimental design) have been included in the materials and methods section (page 15, lines 415-419).

6. Figure 5. Where is Figure 5D-F (page 9, line 215)? D-F is missing from Figure 5 and the Figure 5 legend.  

We apologize for the mistake. We have corrected the error and now data is correctly referred to Figure 5 A-C.

Round 2

Reviewer 1 Report

Please see responses to points 1-4:

1.This reviewer does not see much of difference between 0.1 and 1uM palbociclib (fig 3A). The concern that 1uM is too high still stands.

2. Regardless of the method used, 5uM treatment is too high for T45D cells. The majority of studies utilizing palbociclib in T47D cells use concentrations below 1uM. The concern that the effects shown by palbociclib could be due nonspecific due to the high drug concentration still remains. 

3. The following claim “As expected, all drugs tested reduced cell viability in all cell lines…” is not supported by the data.  The authors have not proved that there is decrease cell viability. It is possible that there is decreased cell proliferation instead and because MTT cannot distinguish the two is not possible to claim decreased cell viability without directly testing cell death. This concern is still valid.

4. While macrophage model shows that SAMHD1 is required for the drug effects observed, different cell lines will show different requirements and thus, the effects shown in Fig 5 could be completely independent of SAMHD1. The concern is still valid.

Round 3

Reviewer 1 Report

Issues were somewhat addressed.